# Whole-genome resequencing reveals collagen-related genes in *Kele pigs*

Yu Dan Zhang[1,2,3☯], Wei Yuan[3☯], Huan Bi[4], Xiao Yang[1,2,3], Yi Yu Zhang[1,2,3], Wei Chen (ID)[1,2,3]*

**1** Key Laboratory of Animal Genetics, Breeding and Reproduction in the Plateau Mountainous Region, Ministry of Education, Guizhou University, Guiyang, Guizhou Province, China, **2** Key Laboratory of Animal Genetics, Breeding and Reproduction, Guiyang, Guizhou Province, China, **3** College of Animal Science, Guizhou University, Guiyang, Guizhou Province, China, **4** Guizhou Agricultural Vocational College, Guiyang, Guizhou, China

☯ These authors contributed equally to this work.
* chenweigzu@163.com

## Abstract

### Objective

To verify the accuracy of collagen-specific SNP mutation loci of *Kele pigs* selected by whole genome resequencing, and to excavate collagen-related genes of *Kele pigs*, so as to lay a foundation for further molecular selection.

### Methods

Based on whole genome resequencing, candidate genes related to collagen trait of *Kele pig* were screened for gene annotation. Through KEGG and GO enrichment analysis of differential genes, we selected four genes that may affect collagen trait of collagen pig, namely *COL9A1*, *COL6A5*, *COL4A3* and *COL4A4*. Then 14 specific SNP sites were randomly selected from the four genes for sanger sequencing verification, and finally RT-qPCR was used to verify the expression levels of related genes in different tissues of *Kele pigs*.

### Results

Our sequencing results revealed that 241.04 G of clean data, Q30 reached 93.96% and the average coverage depth was 9.04×. After data analysis, the SNP annotation of *Kele pigs* identified 4,570 high-impact mutation sites that could result in protein function loss, with SNPs primarily distributed in the intronic and exonic regions. There were 132,256 middle-impact mutation sites and 318,150 low-impact mutation sites that could potentially impact protein properties. Additionally, The INDEL annotation results revealed a total of 17,806 high-impact mutation sites that could potentially result in the loss of protein function. There were 4740 medium-impact mutation sites that have the potential to affect protein properties, as well as 19,298 low-impact mutation sites. Furthermore, there were 14,197,763 mutation sites of modification influence degree in the analysis. In addition, through real-time fluorescence quantitative PCR results, we found that the expression levels of collagen-related genes *COL9A1* and *COL6A5* in skin tissues were higher than those in other tissues, and the

**Data Availability Statement:** All relevant data are within the article and its Supporting Information files.

**Funding:** This work was supported by the Guizhou Science and Technology Plan (Guizhou Kehe

Recruitment [2022] General Rule No. 087), Guizhou University Talent Introduction Research Project (Guida Renji Hezi [2019]34), Guizhou Science and Technology Support Plan (Guizhou Science and Technology Support [2021] General Rule 147) and Guizhou Science and Technology Plan Project (Guizhou Science and Technology Platform Talent [2021]5630).

**Competing interests:** 作者已声明不存在利益竞争。

expression levels of *COL4A4* and *COL4A3* in kidney tissues were higher than those in other tissues. The SNP site verification results showed that the 14 SNP mutation sites randomly selected by us were the same as the SNP mutation sites screened by whole genome resequencing.

## Conclusion

A total of 307 genes related to collagen traits were excavated, including *COL9A1*, *COL6A5*, *EP300*, *SOS2* and *EPO*, etc. It was found that *COL9A1* and *COL6A5* genes were significantly expressed in the skin tissue of *Kele pigs*, and *COL4A4* and *COL4A3* genes were significantly expressed in the kidney tissue of *Kele pigs*. The mutations of 14 randomly selected loci in the four related genes were consistent with the results of previous whole genome resequencing analysis, indicating that the specific SNP molecular marker information obtained by whole genome resequencing can be used as the basis for analyzing collagen traits of *Kele pig*. Our results are conducive to further research on collagen trait regulation of *Kele pigs* and development and utilization of *Kele pigs* in the future.

## Introduction

Collagen constitutes the predominant protein in the animal organism [1], constituting 30% of the total protein content in the body [2]. Collagen contains 18 different types of amino acids, including 7 essential amino acids, making it a highly nutritious substance. As a component of connective tissue in muscles, collagen is widely distributed among muscle fibers and around muscle bundles, forming a fine fibrillar network [3]. Studies have indicated that there is a negative correlation between the tenderness of meat and the content and solubility of muscle soluble collagen. Specifically, a higher content of muscle soluble collagen and greater solubility are associated with lower shear force values, indicating increased tenderness. Therefore, the evaluation of meat tenderness can be achieved through measuring the collagen content in muscle [4]. Therefore, collagen is one of the most important factors affecting meat quality [5]. According to Li Qiannan's research [6], there are several signal transduction pathways associated with collagen, including TGF-β/Smad, PI3K/Akt, MAPK, Wnt, NF-κB, integrin, JAK/STAT, and others. Ricard-Blum [7] demonstrated the discovery of 26 new types of Collagen, including Collagen VIII, in addition to Collagen I, Collagen II, Collagen III, Collagen IV, Collagen V, Collagen VI and Collagen VII. Among them, *COL9A1* gene codes the 1 chain of Type IX collagen and is the main component of Type IX collagen [8, 9]. *COL6A5* gene is a component of type VI collagen [10]. COL6 (collagen VI) is a subtype of collagen found in most connective tissues, consisting of A1 (VI), A2 (VI) and A3, A4 (VI), A5 (VI) and A6 (VI). These six chains are composed of different gene codes (*COL6A1*, *COL6A2*, *COL6A3*, *COL6A4*, *COL6A5*, *COL6A6*) [11]. *COL4A4* and *COL4A3* genes encode the a3 and a4 chains of type IV collagen, which are the components of type IV collagen [12]. The *Kele pig* is considered one of the top pig breeds in China. It originates from Hezhang, Guizhou, and can be found in the northwest of Guizhou, as well as in Xuanwei and Qujing of Yunnan. This breed is recognized as a valuable local genetic resource in Guizhou and has been documented in the Annals of 《Chinese Livestock and Poultry Genetic Resources · Pig Annals》 [13–15]. Being a characteristic local pig breed in Guizhou, the *Kele pig* is known for its fine meat quality and excellent flavor [16]. Hence, it is of significant importance to investigate the genes associated with the regulation of

collagen-protein characteristics in *Kele pigs*. Based on whole genome resequencing, this study conducted a screening of collagen-regulating differential genes in Collagen-regulating pigs. The study also explored the expression levels of four genes (*COL9A1*, *COL6A5*, *COL4A4* and *COL4A3*) in seven different tissues of Collagen-regulating pigs: skin, longissimus dorsi muscle, heart, liver, spleen, lung and kidney. This research lays a foundation for further exploration into the regulation of collagen-regulating traits in Collagen-regulating pigs.

## Experimental samples and methods

### Experimental samples

This research group collected skin, longissimus dorsi muscle, heart, liver, spleen, lung, kidney and other tissues of *Kele pigs* at the *Kele pig* breeding farm in Hezhang County, Bijie City, Guizhou Province. The tissue samples were initially treated with DEPC water and then frozen in liquid nitrogen using sterile enzyme-free freeze-storage tubes. Subsequently, they were transferred to an ultra-low temperature refrigerator at -80°C for DNA and RNA extraction.

## Experimental methods

### Extraction of genomic DNA and construction of library

In this experiment, the phenol-chloroform method was utilized for DNA extraction. The specific procedures were as follows: ear tissue samples were collected and cut into pieces, then added to a centrifuge tube containing lysis buffer. The DNA was purified using an agarose column, and impurities such as protein and RNA were subsequently removed. Next, DNA extraction was carried out using phenol/chloroform, followed by DNA precipitation with ethanol. Subsequently, impurities were removed using a wash buffer and the genomic DNA was dissolved in TE buffer. DNA concentration and A260/280 values were measured, and TE buffer was added as necessary to adjust the concentration to a range of 50-300ng/ul. Subsequently, agarose gel electrophoresis was conducted to assess the quality of the DNA samples. The genomic DNA is fragmented using sonication, followed by chemical end repair and ligation to Illumina sequencing primers. The resulting DNA fragments are then amplified by PCR. Subsequently, the amplified DNA fragments are subjected to gel electrophoresis for screening, with only the target length DNA fragments being retained and purified. Then, the purified DNA fragments are inserted into the vector, and the library is transformed into *E. coli* by electrotransformation. Following screening and amplification, the library is constructed. The resulting genome library can be utilized for high-throughput sequencing in order to obtain comprehensive organism whole genome sequence information.

### Variation detection and filtering based on whole genome resequencing

Wu Han gene read Technology Co., LTD was commissioned to complete the whole genome resequencing of *Kele pig*. The main sequencing platform was Illumina HiSeq 2500, the sequencing strategy was PE125, and the depth was 10 x. The Fsatp v0.19.10 software was used to filter the original sequencing data, remove adaptor and low-quality sequences, and compare the filtered data to the porcine reference genome Sscrofa11.1 using bwa v0.7.17-r1188 software. The sequenced data was then de-duplicated to obtain a bam file for mutation detection. The read was compared to the pig reference genome (Sscrofa11.1) downloaded from the NCBI database using samtools vl.16.1 and sorted based on the length score of the comparison. Once sorting is complete, Picard MarkDuplicates v2.18.29-SNAPSHOT is utilized to eliminate duplicate comparisons. Use GATK v4.0 (https://gatk.broadinstitute.org/hc/en-us) Haplotype-Caller tools to respectively analyze the *Kele pig* bam files for mutation detection. The gvcf file

for each pig was then merged into a single population gvcf file containing variation across the entire population. Filter SNP and INDEL using the VariantFiltration tool in GATK 4.0. The quality control filter parameters of SNPs and INDELs are QD < 2.0 || MQ < 40.0 || FS > 60.0 || SOR > 3.0 || MQRankSum < -12.5 || ReadPosRankSum < -8.0.

## Assessment of effects of SNPs and INDELs along with extraction of their associated genes

The variations were annotated using snpeff v5.0e software, and all SNPs were categorized into exons, introns, gene upstream regions, splicing sites, and other genomic regions based on their positions. SNPs classified into exon regions are further categorized as either synonymous variants or non-synonymous variants. The annotation process for INDELs is similar to that of SNPs, with the distinction between insertion and deletion regions based on their location in the genome. The body parameter is download-c./snpEff.config -v ebola_zaire Sscrofa11.1.99, -c./ snpEff.config-ud 2000-csvStats-htmlStats ". The annotation operation of INDEL is the same as that of SNPS, which are divided into insertion or deletion regions according to their location on the genome.

## GO and KEGG analyses of genes associated with SNPs and INDELs

KEGG uses the R package clusterProfiler v4.0 to call the official website of KEGG database (https://www.kegg.jp/)API) directly for KEGG analysis, and uses the built-in function of the R package dotplot for visual analysis. GO via R package org.Ss.eg.dbv3.16.0(http://bioconductor. org/packages/release/BiocViews.html#___OrgDb) is analyzed, and the use clusterProfiler built-in function dotplot visualization analysis; Finally through the online website and kish became the father of letter cloud tools dior (https://www.omicshare.com/tools/Home/Soft/ cog) into the line of visualization.

## PCR based validation of 14 SNPs and their associated genes

In order to verify the specific SNP sites of *Kele pigs*, 16 sites were randomly selected from the previously screened specific SNPs sites of *Kele pig* (Table 1) for verification. We used genomic DNA extraction kit (Thermo GeneJET) to extract DNA from ear tissue samples of 50 *Kele pigs*. According to the sequences of 100 bp before and after SNP sites were selected, specific primers were designed using Primer Premier 5.0 software (Table 2). The primers were synthesized by Qingke Biotechnology Co., LTD. PCR amplification system 50 μL: DNA template 2μL, upper and downstream primers 2μL, 2×EasyTaqPCR SuperMix 25 μL, ddH2O 19 μL. PCR reaction procedure: predenaturation at 94°C for 2 min; Denatured at 94°C for 30 s, annealed (annealing temperature is shown in Table 2) for 30 s, extended at 72°C for 30 s, a total of 33 cycles, extended at 72°C for 2 min, stored at 4°C. Then 1.0% agarose gel electrophoresis was used for detection. Finally, the PCR products identified correctly were sent to Qingke Biotechnology Co., Ltd. for sanger sequencing.

## RT-fluorescent qPCR based validation of role of SNP associated genes in collagen expression

**RNA extraction and cDNA synthesis.** In this experiment, the Trizol method was utilized for RNA extraction. The specific steps were as follows: skin, longisbest dorsi muscle, heart, liver, spleen, lung and kidney samples were collected and fully ground. Subsequently, they were added into a centrifuge tube containing Trizol. Chloroform was then added to remove the protein followed by centrifugation and absorption of the upper liquid into a new centrifuge

**Table 1. Information of specific SNPs sites.**

| Gene | SNPs | REF | ALT | SITE |
|---|---|---|---|---|
| COL9A1 | SNP1 | G | C | g.50509272 |
| | SNP2 | G | T | g.50509314 |
| | SNP3 | C | T | g.50509226 |
| COL6A5 | SNP4 | T | C | g.2117893 |
| | SNP5 | T | G | g.2117929 |
| | SNP6 | C | T | g.2118081 |
| COL4A4 | SNP7 | C | G | g.128531420 |
| | SNP8 | C | T | g.128531424 |
| | SNP9 | G | A | g.128531440 |
| | SNP10 | T | G | g.128531456 |
| COLA43 | SNP11 | G | A | g.84423643 |
| | SNP12 | C | T | g.84423793 |
| | SNP13 | G | A | g.84423810 |
| | SNP14 | C | A | g.84423811 |

" REF " refers to genomic locus information; "ALT " mutation site information; " SITE " Location of the mutation site.

tube. Next, isopropyl alcohol was added to the new centrifuge tube to precipitate the RNA. The mixture was then centrifuged and the supernatant discarded. Subsequently, 75% anhydrous ethanol was added to wash the RNA. After drying, the RNA was dissolved in DEPC water. The purity of the RNA was evaluated by determining its concentration and 260/280 values. The extracted RNA was then reverse-transcribed into cDNA and stored at -80°C for future use.

**Primer design and synthesis.** According to the sequences of the *COL9A1* gene (NC_010443.5), *COL6A5* gene (NC_010455.5), *COL4A4* gene (NC_010457.5) and *COL4A3* gene (NC_010457.5) in the GenBank database, we used Premier 5.0 software to design one

**Table 2. Specific primer information.**

| Gene | Primers sequence | Product length (bp) | Annealing temperature (°C) |
|---|---|---|---|
| COL9A1 | F: TGGATTGCCAAATCAGACACTT | 136 | 59 |
| | R: GGGAAGCATTTCTGGTGGTCT | | 61 |
| | F: ACCACCAGAAATGCTTCCCA | 188 | 60 |
| | R: CATGGCCTGTTAGAAAGTGTCA | | 59 |
| | F: TGATTACCGTGAGACACAGAAGG | 134 | 60 |
| | R: TGCCCACATGTGCTACTTTG | | 59 |
| COL4A4 | F: CAAGAGTTAAAGGGCACAAAG | 95 | 55 |
| | R: AGCATACCCATCTCGACCA | | 58 |
| COL6A5 | F:GGCTGGACGATGTAACTGT | 129 | 57 |
| | R:AATTTCCCAAGTCTGCATAGGATTC | | 59 |
| | F:GTGTCCCACATCACCCAGATT | 170 | 60 |
| | R:ATCTAAGAGTGTCTGTAATAGCTGG | | 58 |
| COL4A3 | F:CCAGGCTCTTGCTTGGAAA | 137 | 58 |
| | R:ATAGGTTTCCTGAACATTCTTTCGG | | 59 |
| | F:ACCAGTGTCAGGCACATAGG | 144 | 59 |
| | R:TCTGAAGGAATTTGAAAAGAGGAAG | | 57 |

**Table 3. Primers sequence.**

| Genes | Primers sequence | Product lengthk (bp) | Annealing temperature (°C) |
|---|---|---|---|
| Col9a1 | F:GCAAGTTGGCGTGAAGATA | 120 | 56 |
| | R:CACTGGGAATCAAACAAGGAGGGCA | | 65 |
| Col6a5 | F:GGCTGGACGATGTAACTGT | 129 | 57 |
| | R:AATTTCCCAAGTCTGCATAGGATTC | | 59 |
| Col4a4 | F:CAAGAGTTAAAGGGCACAAAG | 92 | 55 |
| | R:AGCATACCCATCTCGACCA | | 58 |
| Col4a3 | F:CCAGGCTCTTGCTTGGAAA | 137 | 58 |
| | R:ATAGGTTTCCTGAACATTCTTTCGG | | 59 |

pair of real-time fluorescent quantitative PCR primers for each gene. The primers were synthesized by Qing Ke Biotechnology Co., LTD, and their sequences are shown in Table 3.

**Real-time fluorescence quantitative PCR detection.** Using the cDNA from each tissue as a template, ABI QuantStudioTM7 Flex real-time fluorescence quantitative PCR was employed for detection. The reaction system consisted of 10 μL: 0.2 μL of 10 μmol/L forward and reverse primers, 1 μL of cDNA, 5 μL of 2xEs Taq MasterMix (Dye), with water added to reach a total volume of 10 μL. The reaction conditions included an initial denaturation at 95°C for 5 min; followed by denaturation at 95°C for 10 s, annealing at 56°C for 30 s, extension at 72°C for 30 s, and the reaction proceeded through 45 cycles.

## Results

### DNA quality testing of samples of *Kele pig*

Fig 1 illustrates that the extracted genomic DNA bands of *Kele pig* are singular and clear, with a high concentration. Additionally, the DNA from all 5 samples meet the requirements for

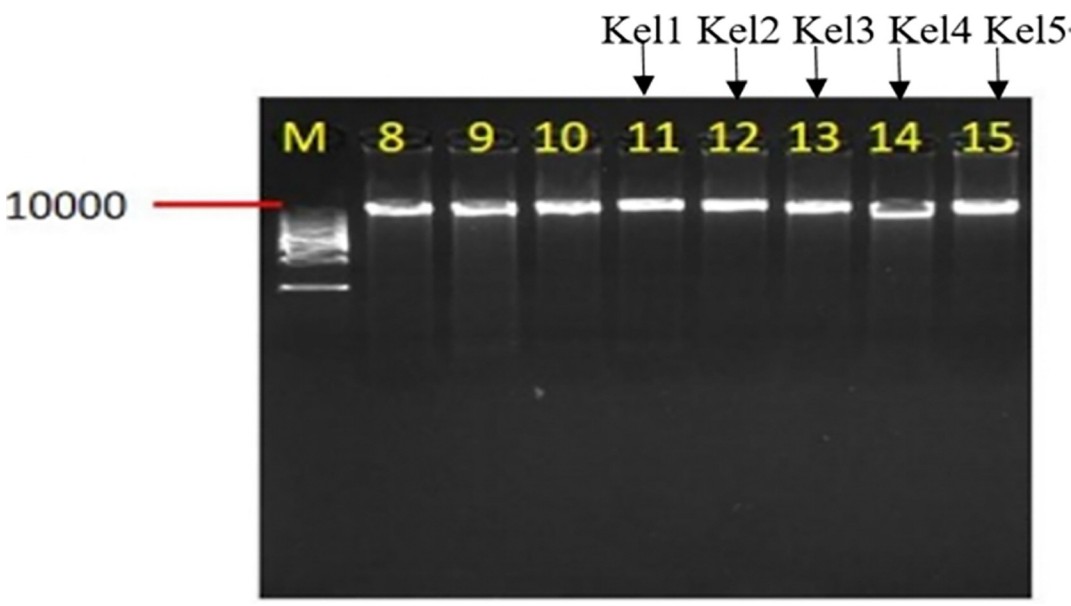

**Fig 1. DNA quality test results of *Kele pigs*.**

**Table 4. Results of DNA OD value of *Kele pig*.**

| Serial number | Sample name | Concentration (ug/ml) | OD260/280 | OD260/230 |
|---|---|---|---|---|
| 1 | Kel1 | 55.6 | 1.85 | 1.14 |
| 2 | Kel2 | 378 | 1.91 | 1.96 |
| 3 | Kel3 | 442 | 1.93 | 1.86 |
| 4 | Kel4 | 812 | 1.89 | 1.5 |
| 5 | Kel5 | 580 | 1.97 | 1.94 |

sequencing in the database. Table 4 indicates that the OD260/OD280 values of the extracted genomic DNA from *Kele pig* range between 1.8 and 2, indicating good quality DNA samples.

### Sequencing data quality analysis

After rigorous filtering of the sequencing data, high-quality clean data was obtained. The output data underwent statistical analysis (Table 5), which included sequencing data output, sequencing error rate, Q20 content, Q30 content, GC content, and other relevant parameters. The five samples collectively generated a total of 241.8 G of raw data, averaging 48.36 G of raw data per sample. Following filtration, the clean data totaled 241.04 G with an average of 48.2 G per sample. The Q30 of the filtered samples were all higher than 90%, which reached the quality control qualified data standard, and the sequencing data could be used for the next analysis.

### Compare with the reference genome

The filtered clean data was compared to the pig reference genome (Sscrofa11.1). The comparison results indicated that the comparison rate between each individual and the reference genome ranged from 93% to 98%. Additionally, the average sequencing depth of *Kele pig* samples was determined to be 9.04×. For the ratio index of five individuals to the reference genome, it is evident that the average ratio of this index exceeds 90%, which satisfies the criteria for resequencing analysis. Consequently, the data obtained from the current resequencing analysis meets the requirements for subsequent analysis.

### Detection results of SNPs and INDELs in *Kele pigs*

According to Table 6, a total of 50,040,8.74 million SNPs loci and 12,037,063 SNPS in gene regions were obtained from samples Kel1, Kel2, Kel3, Kel4 and Kel5. The majority of the SNPS were found to be distributed in the intronic and exonic regions. Additionally, 14,216,261 INDELs loci and 3,375,354 gene region INDELs loci were identified. A total of 7146.2 million SNPs and 99,992 INDELs were identified within the exon region.

**Table 5. Quality results of sequencing data.**

| Sample name | Sample abbreviation | Raw data(G) | Filter number (G) | Q20 (%) | Q30 (%) | GC (%) |
|---|---|---|---|---|---|---|
| Kele-1 | kel1 | 40.27 | 40.11 | 98.19 | 93.61 | 41.47 |
| Kele-2 | kel2 | 38.1 | 37.98 | 98.22 | 93.7 | 41.25 |
| Kele-3 | kel3 | 50.45 | 50.3 | 98.29 | 93.97 | 41.78 |
| Kele-4 | kel4 | 63.1 | 62.9 | 98.42 | 95.15 | 42.17 |
| Kele-5 | kel5 | 49.88 | 49.75 | 98.29 | 93.39 | 41.36 |

"Q20" The percentage of bases with a mass value ≥20; "Q30" The percentage of bases with a mass value ≥30; "G" Data unit.

**Table 6. Detection results of SNPs and INDEL in *Kele pig* samples.**

| Variation region | SNP | | INDEL |
|---|---|---|---|
| | *Kele pig* | | |
| exon | 714,620 | | 99,992 |
| Gene interval | 12,037,063 | | 3,375,354 |
| intron | 34,308,963 | | 9,869,994 |
| shear | 1,257 | | 927 |
| upstream | 1,114,108 | | 336,723 |
| downstream | 1,193,411 | | 354,421 |
| 3 'end | 506,819 | | 146,995 |
| 5 'end | 164,633 | | 31,855 |
| total | 50,040,874 | | 14,216,261 |

## SNPs annotation results of *Kele pigs*

The SNPs classified by their degree of influence in the annotation results were tallied. The analysis revealed 4570 mutation sites that could result in loss of protein function, as well as 132,256 mutation sites that could impact protein properties in *Kele pigs*. Additionally, there were 318,150 low-impact mutation sites identified in *Kele pigs*. Detailed results can be found in Table 7.

The SNPs classified by gene mutation function in the annotation results were subjected to statistical analysis. A total of 24,971,814 SNPs were annotated, with 246,426 being synonymous mutations, accounting for 64.67% of the functional mutations. Non-synonymous mutations totaled 134,584, representing 35.33% of the functional classes of gene mutations. Within the non-synonymous mutations category, there were 132,872 missense mutations that altered the coding amino acid codon (34.87% of the functional categories). Additionally, there were 1,712 nonsense mutations resulting in stop codons after point mutations and prematurely terminating peptide synthesis (0.44% of the functional types). Detailed results can be found in Table 8.

As illustrated in Fig 2, there were a total of 61,012,906 mutation sites within nucleotides of the same class and 25,399,662 mutation sites between nucleotides of different classes in *Kele pigs*. The Ts/Tv ratio was calculated to be 2.4021, indicating a significantly higher prevalence of base conversion type mutations compared to transmutation type mutations.

**Table 7. Presents the SNP annotation results of *Kele pigs*, classified according to their degree of influence.**

| Functional mutation type | *Kele pig* | |
|---|---|---|
| | quantity | Percent (%) |
| High impact | 4570 | 0.009 |
| Medium impact | 132,256 | 0.264 |
| Low impact | 318,150 | 0.635 |
| Modification effect | 49,642,823 | 99.092 |

**Table 8. SNPs annotation results classified by mutation function in *Kele pigs*.**

| Functional mutation type | *Kele pig* | |
|---|---|---|
| | quantity | Percent (%) |
| Missense mutation | 132,872 | 34.874 |
| Nonsense mutation | 1,712 | 0.449 |
| Synonymous mutation | 246,426 | 64.677 |

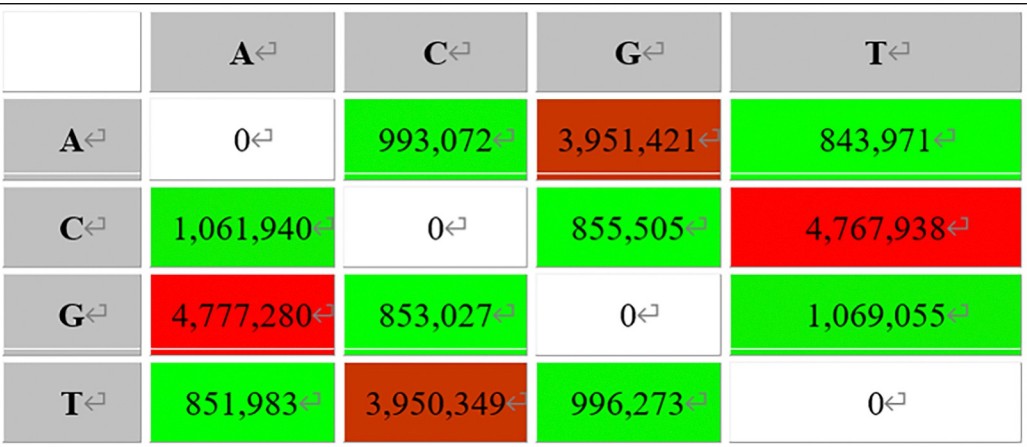

**Fig 2. Base changes of SNPs variation in *Kele pigs*.**

## INDELs annotation results of *Kele pigs*

The number of mutation sites influencing mutant genes in the annotation results was calculated. Specifically, there were 17,806 mutation sites that could result in the loss of protein function and 4740 mutation sites that could impact protein properties in *Kele pigs*. Additionally, there were 19,298 mutation sites with a low impact degree in *Kele pigs* and 14,197,763 mutation sites with a degree of modification. Detailed results can be found in Table 9.

Statistics were conducted on the insertion and deletion marks in the INDEL annotation results. A total of 7,041,172 insertion and deletion marks were identified in *Kele pig*, comprising 4,031,333 insertion marks and 3,009,839 deletion marks. The detailed findings are presented in Table 10.

## Gene enrichment analysis of the selected region of the *Kele pig* population

KEGG enrichment analysis was conducted on genes within the selected regions. The results of the KEGG enrichment analysis revealed that relevant collagen pathways were enriched in six pathways, specifically PI3K/Akt, Jak STAT signaling pathway, MAPK, Wnt, NF-κB, and

**Table 9. Results of INDELs annotations classified by impact degree in *Kele pigs*.**

| Functional mutation type | Kele pig | |
|---|---|---|
| | quantity | Percent (%) |
| High impact | 17,806 | 0.125 |
| Medium impact | 4740 | 0.033 |
| Low impact | 19,298 | 0.136 |
| Modification effect | 14,197,763 | 99.706 |

**Table 10. Results of INDELs annotation by mutation type.**

| Mutation type | Kele pig | |
|---|---|---|
| | quantity | Percent (%) |
| interposition | 4,031,333 | 42.75 |
| deficiency | 3,009,839 | 57.25 |

**Table 11. Part of KEGG enrichment pathways and genes regulating collagen.**

| Serial number | KEGG pathway | Gene |
|---|---|---|
| 1 | PI3K-Akt signaling pathway | *EGF,SPP1,FLT3,COL4A2,MAPK1, COL4A4,COL4A3,COL9A1,COL6A5* |
| 2 | Jak-STAT signaling pathway | *AKT2,CREBBP,EP300,OSMR,PIK3R1, PDGFRA,PDGFRB,PRL,PRLR* |
| 3 | MAPK signaling pathway | *AKT2,CD14,EPHA2,FOS,KITLG, KIT,MAPKAPK3,MKNK2,MECOM* |
| 4 | Wnt signaling pathway | *EP300,PPARD,DAAM2,NFATC4,DVL1, ROR1,PLCB2,INVS,RSPO2* |
| 5 | TGF-beta signaling pathway | *FBN1,THSD4,BMP4,TGFBR1,RBL1, CREBBP,MAPK1,TGFB2,RPS6KB2* |
| 6 | NF-kappa B signaling pathway | *TAB1,LTBR,IRAK4,RIPK1,PLCG2, LCK,TNFAIP3,TNFRSF11A,TLR4* |

transforming growth factor-β signaling pathway. A total of 475 candidate genes with pathway annotations were identified (Table 11). GO enrichment analysis was conducted on the genes within the selected regions, resulting in a total of 30 pathways related to collagen. The GO enrichment and KEGG enrichment results are presented in Figs 3 and 4, respectively. Notably, the collagen-related genes *COL9A1* (NC_010443.5), *COL6A5* (NC_010455.5), *COL4A4* (NC_010457.5), and *COL4A3* (NC_010457.5) were identified in both GO and KEGG

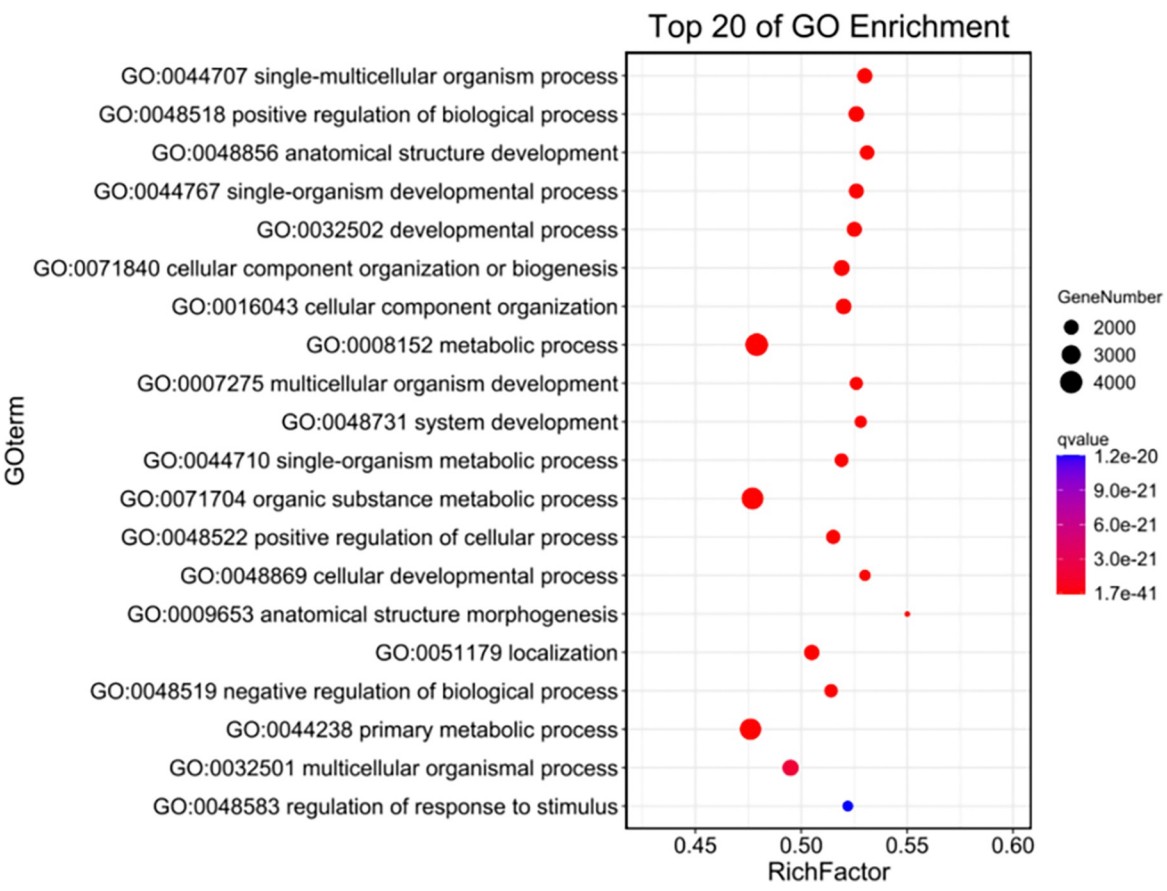

**Fig 3. Gene GO enrichment in selected regions of *Kele pig* population.**

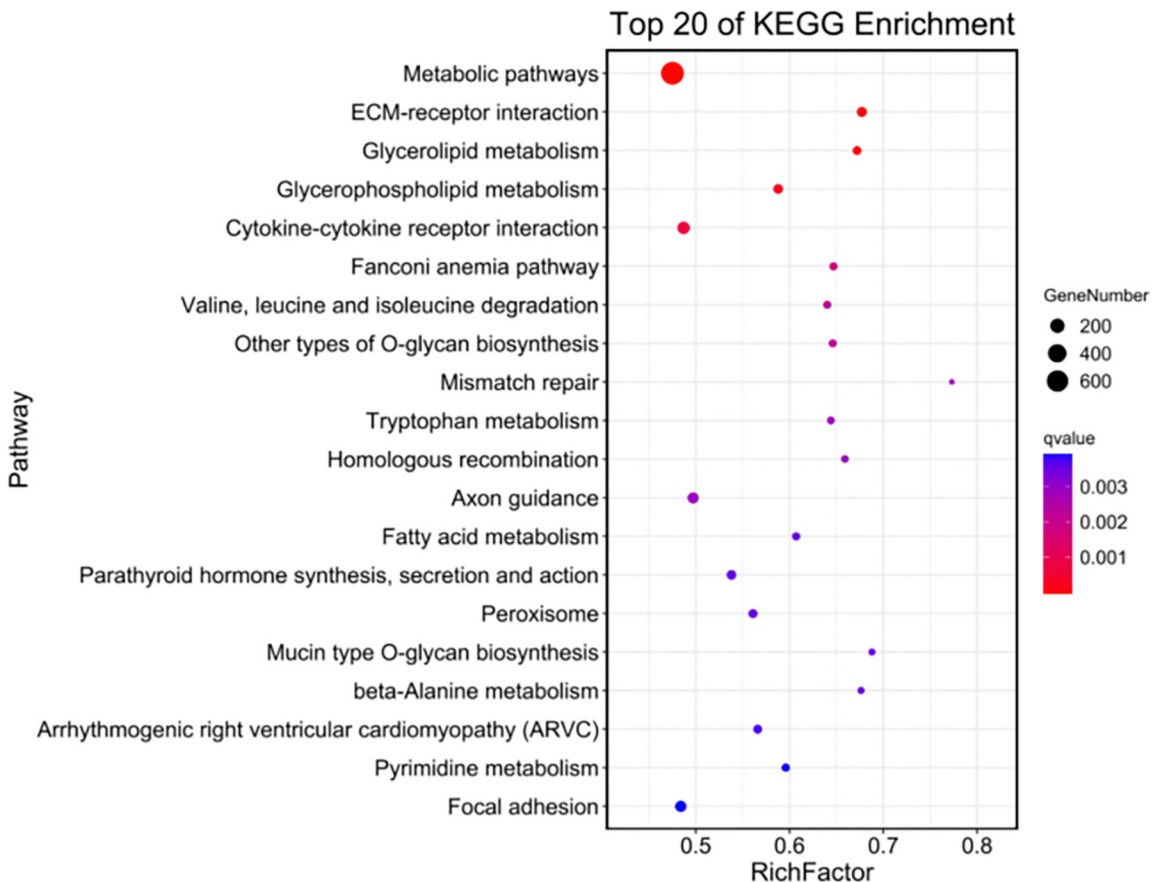

**Fig 4. KEGG gene enrichment in selected regions of *Kele pig* population.**

enrichment analyses. These findings suggest that these genes could be considered as potential candidates for encoding collagen proteins in further studies.

## SNP sites validation

14 validated sites were randomly selected, That is, G. 50509272G > C, G. 50509314G > T, G. 50509226C > T, G. 2117893T > C, G. 2117929T > G, G. 2118081C > T, G. 128531420C > G, G. 128531424C > T, G. 1285 31440G > A, G.128531456T > G, G.84423643G > A, G.84423793C > T, G.84423810g > A, G.84423811C > A, and the specificity of the 14 SNPs sites was the same as that of the SNP mutation sites screened by whole genome resequencing. The verified SNPS were found only in a specific population, the results were as expected (Figs 5–8).

## The expression of related genes in different tissues

The expression levels of *COL9A1*, *COL6A5*, *COL4A4*, and *COL4A3* in skin, muscle, heart, liver, spleen, lung, and kidney tissues of *Kele pigs* were investigated using RT-qPCR. The results indicated that the expression levels of *COL4A4* and *COL4A3* in *Kele pig* kidney tissues were higher than those in other tissues. Additionally, the expression of genes *COL9A1* and *COL6A5* in the skin tissue of *Kele pigs* was also found to be higher than that in other tissues (Fig 9).

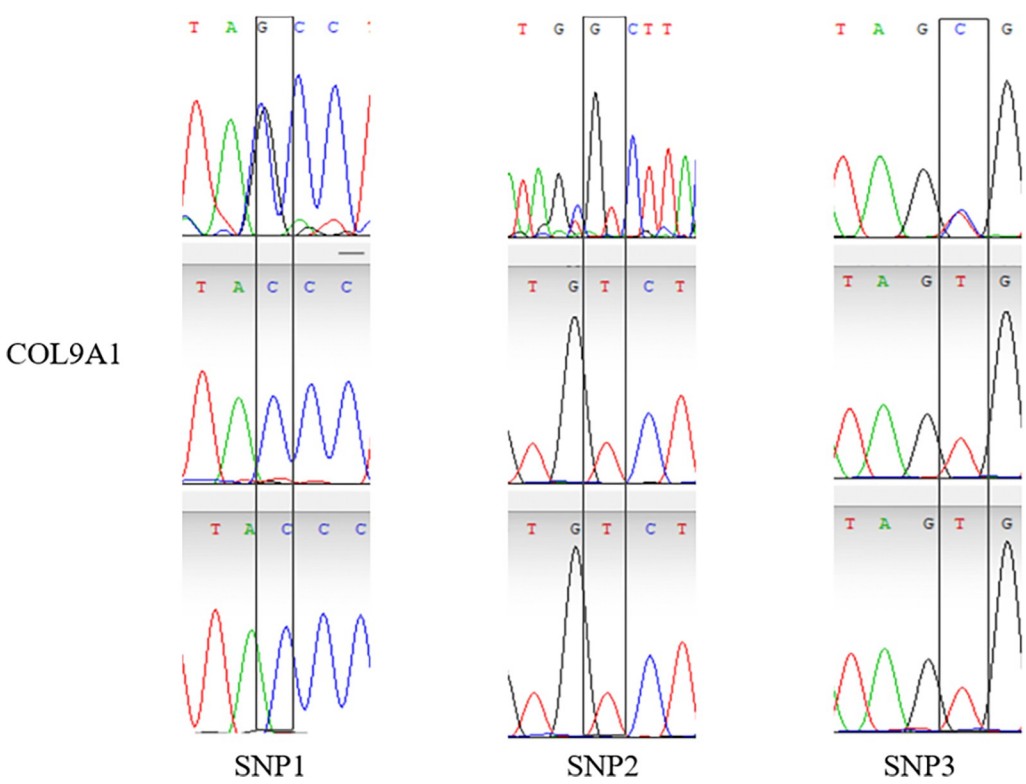

**Fig 5. Peak map of SNPs locus with variation in Sanger sequencing of *COL9A1* gene.**

## Discussion

Based on the sequencing data and reference genome, a total of 50,040,8.74 million SNPs loci and 14,216,261 INDELs loci were identified. The variations were predominantly found to be distributed in intergenic and intronic regions in this study. Intergenic variation may impact gene expression, RNA post-transcriptional modification, transcription factor binding, and splicing. Variation within introns may influence the occurrence of splicing events, thereby impacting the amino acid sequence that encodes for proteins and ultimately the structure and function of proteins. Ryu et al. [17] identified stroke-related SNPs in the gene spacer region between *FOXF2* and *FOXQ1* in zebrafish. These SNPs were found to regulate the enhancer activity and expression of the vascular stability regulator *FOXF2*, thus playing a role in regulating stroke risk in both human cells and zebrafish. High-impact mutations can lead to the loss of protein function, while medium-impact mutations may impact the performance of the protein. Low-impact mutations are unlikely to affect protein function, and modified variants have no direct effect on gene and protein function. The number of high-impact mutation sites in *Kele pigs* was 17,806, which can be highly disruptive to gene or protein function. In the context of functional mutation types of SNPs, non-synonymous mutations have the potential to alter the amino acid sequence, thereby impacting the structure and function of the protein. According to the findings of this study, a total of 134,584 non-synonymous mutations were identified in *Kele pigs*. In addition, base translocations can also result in significant changes in gene function, where a single base is substituted by another in a DNA sequence mutation. If both of the substituted bases are purines or pyrimidines, it is referred to as Transition (Ts); if one of the two replaced bases is a purine and the other is a pyrimidine, it is known as Transversion (Tv).

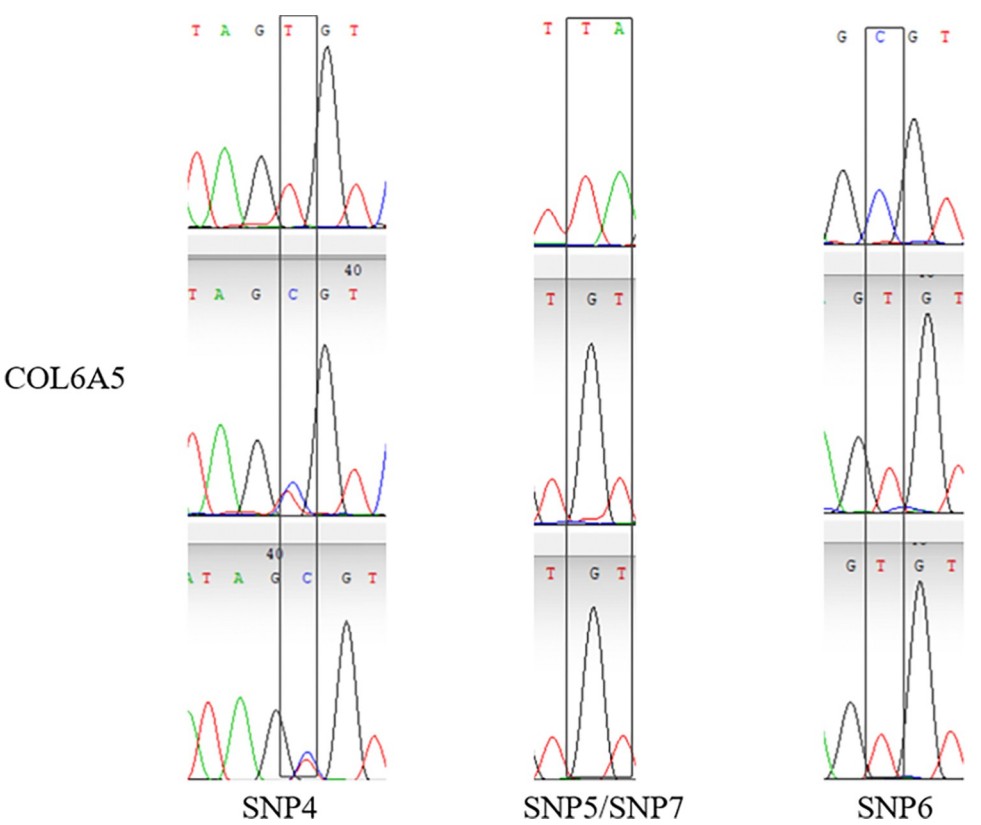

**Fig 6. Peak map of SNPs locus with variation in Sanger sequencing of *COL6A5* gene.**

This study identified 25,399,662 transversion points in *Kele pigs*. Liu et al. [18] demonstrated that a single non-synonymous mutation in the E-protein coding sequence 9 of the Zika virus can significantly increase neurovirulence in vivo. Similarly, Matsumoto et al. [19] found that non-synonymous mutations in the bovine *SPP1* gene may impact muscle development, leading to an increase in carcass weight of C/T animals. Therefore, non-synonymous mutations can alter protein structure and function for both beneficial and detrimental effects. Further investigation of the high-impact mutation sites, non-synonymous mutation sites, and translocation sites identified in this study can provide a better understanding of gene function and regulatory mechanisms in *Kele pigs*. INDEL refers to the insertion or deletion of a base in the DNA sequence. These variations can impact the function, structure, and genetic characteristics of the genome, playing a crucial role in genetic breeding exploration. Niu et al. [20] discovered that inserting 12 bases into the 3'UTR region of the *CISH* gene in Landwhite pigs could affect the susceptibility of landwhite piglets to diarrhea. Mi et al. [21] also found that three INDEL variation sites (L-13, L-16, and L-19) on the gene of cilia and flagella-related protein *CFAP43* were significantly correlated with growth traits such as chest depth in goats. In this study, a total of 7,041,172 insertion-deletion markers and 17,806 high-impact mutation sites were identified in *Kele pigs*. The discovery of these INDEL variation sites holds significant importance for comprehending the structure and function of the *Kele pig* genome. Furthermore, it provides valuable information for gene editing and genetic modification. We used sanger sequencing technology to verify the specific loci of *Kele pigs* previously screened. This verification method has also been reported in other studies. For example, Zhu Tao et al. [22] selected a total of 56 ducks from 7 populations, namely Cherry Valley Duck, Beijing duck, Maple Leaf

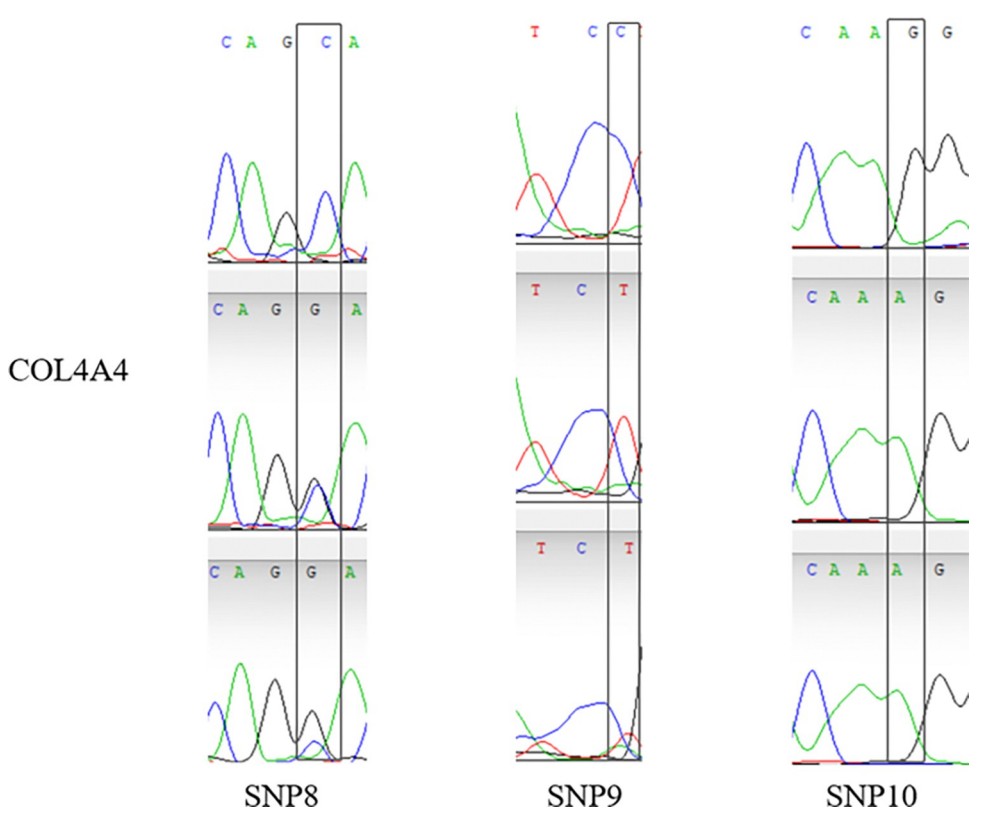

**Fig 7. Peak map of SNPs locus with variation in Sanger sequencing of *COL4A4* gene.**

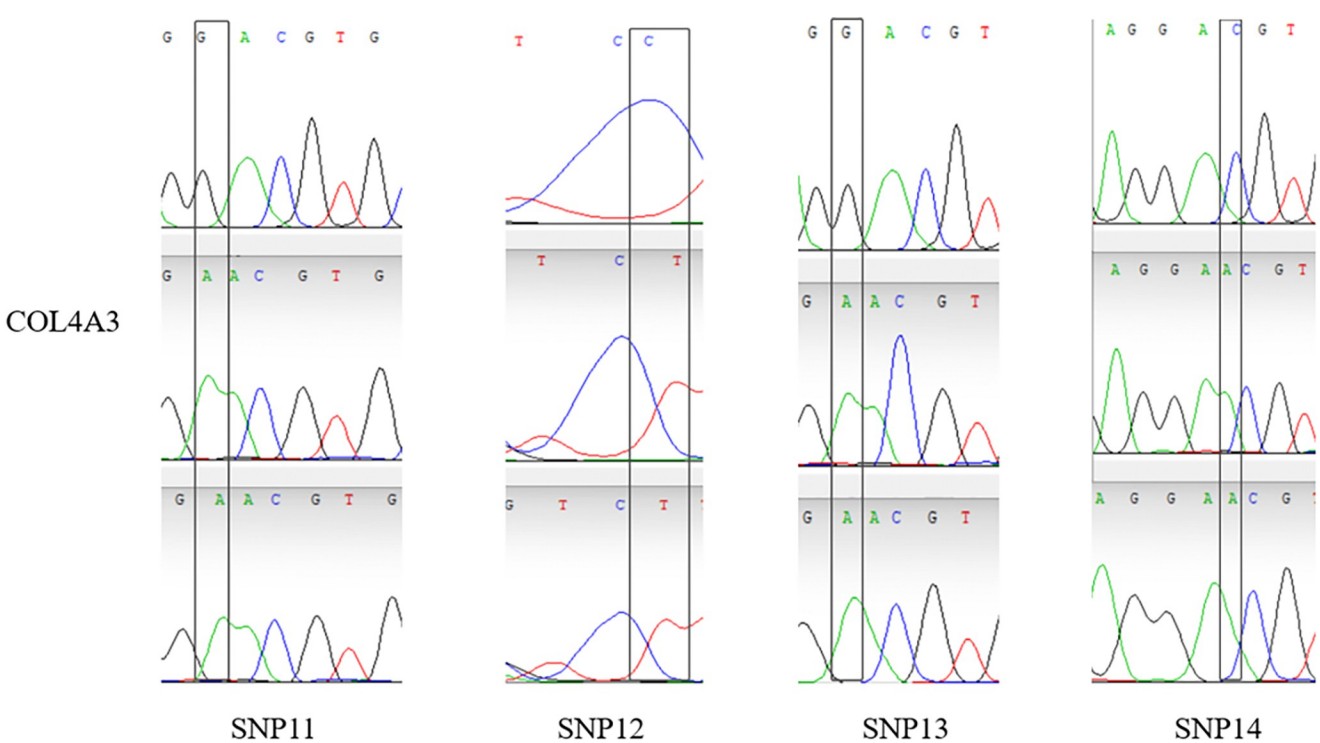

**Fig 8. Peak map of SNPs locus with variation in Sanger sequencing of *COL4A3* gene.**

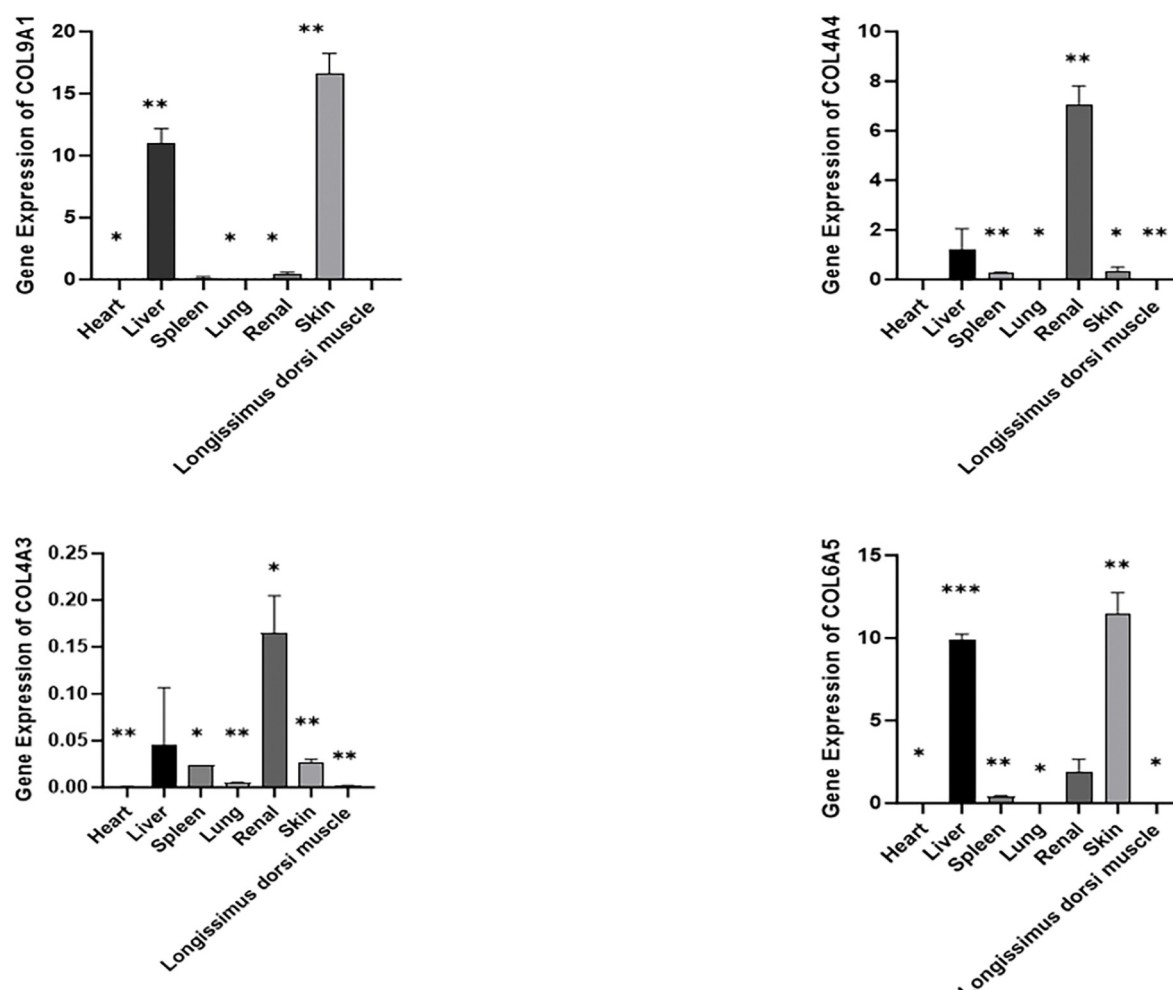

**Fig 9. Expression levels of *COL9A1*, *COL6A5*, *COL4A4* and *COL4A3* genes in different tissues of *Kele pigs*.** "*"Significant difference (p<0.05),"**"The difference is extremely significant (p<0.01).

duck, Jinding duck, Shaoxing duck, Shanma duck and Gaoyou duck, and conducted whole genome resequencing. 686 SNPs and Indel existing in specific populations were screened, and 7 SNPs sites were randomly selected for verification. The results were consistent with expectations. The specific SNPS screened in the second generation sequencing could be used as molecular markers for the identification of duck breeds. Fan Huanhuan et al. [23] randomly selected 30 red deer specific SNPs sites for sanger sequencing based on genotyping sequencing technology, and the research results indicated that red deer specific SNPs screened by GBS sequencing could be used as molecular markers for identification. This study is based on whole genome sequencing. The pathways involved in the regulation of collagen were screened, including the PI3K-Akt signaling pathway, Jak-STAT signaling pathway, MAPK signaling pathway, Wnt signaling pathway, TGF-beta signaling pathway, NF-kappa B signaling pathway, etc. A total of 133 genes were identified in the PI3K-Akt signaling pathway to regulate collagen-related traits in *Kele pigs*. Whole gene resequencing and GO analysis were utilized to investigate the genes associated with collagen deposition in *Kele pigs*, resulting in the identification of a total of 307 major candidate genes. The results revealed that these related genes were significantly enriched in several signaling pathways, including the PI3K-Akt signaling

pathway, Jak-STAT signaling pathway, MAPK signaling pathway, Wnt signaling pathway, TGF-beta signaling pathway, and NF-kappa B signaling pathway. The PI3K-Akt signaling pathway regulates a variety of cellular processes in response to extracellular signals, including metabolism, proliferation, cell survival, growth, and angiogenesis [24]. The MAPK signaling pathway plays a critical role in cell proliferation, differentiation, apoptosis, and metabolism [25]. Hu et al. [26] showed that LPS promoted collagen synthesis in lung fibroblasts by activating the PI3K-Akt-mTOR/PFKFB3 pathway and aerobic glycolysis. Therefore, from the PI3K-Akt signaling pathway, *COL9A1*, *COL6A5*, *COL4A4* and *COL4A3* genes related to collagen-deposition in *Kele pigs* were preliminatively screened. In recent years, there have been many studies on the correlation between *COL9A1* and Kashin-beck disease [27], congenital clubfoot [28] and tumors. Currently, *COL6A5* gene has been associated with lipid metabolism [29], proliferation and angiogenesis of colon cancer cells [30]. The study of Miner et al. [31]. showed that gene mutations encoded by *COL4A3* and *COL4A4* could lead to glomerular diseases in humans and mice.

According to literature, the protein content of pig skin is reported to be as high as 33%, with collagen accounting for 87.7% of this total [32]. In summary, this study randomly selected 14 SNP-related variation sites through whole genome resequencing analysis and conducted sanger sequencing to verify that the results were in line with expectations. That is, G. 50509272G > C, G. 50509314G > T, G. 50509226C > T, G. 2117893T > C, G. 2117929T > G, G. 2118081C > T, G. 128531420C > G, G. 128531424C > T, G. 1285 31440G > A, G.128531456T > G, G.84423643G > A, G.84423793C > T, G.84423810G > A, G.84423811C > A These mutation sites can be used as a reference for marker assisted selection. We initially explored the genes closely related to collagen-protein traits of *Kele pigs*, and detected the expression of genes *COL9A1*, *COL6A5*, *COL4A4* and *COL4A3* regulating collagen-protein traits in the heart, liver, spleen, lung, kidney, longissimus dorsi muscle and skin tissues of *Kele pigs* by qRT-PCR method. The results showed that both *COL9A1* and *COL6A5* genes were significantly expressed in the skin tissue of *Kele pigs*, and both *COL4A4* and *COL4A3* genes were significantly expressed in the kidney tissue of *Kele pigs*. The study laid the groundwork for further investigation into the regulation mechanism of collagens by *COL9A1*, *COL6A5*, *COL4A4*, and *COL4A3* genes. Based on the expression patterns of these genes in the heart, liver, spleen, lung, kidney, skin, and longissimus dorsi muscle of *Kele pigs*, it was found that *COL9A1* and *COL6A5* genes were significantly expressed in the skin of *Kele pigs*. This suggests a need for deeper exploration into how these genes regulate collagenic protein traits in Collagenic pigs. The findings provide a scientific basis for future research on the regulatory mechanisms of collagenic proteins related to Collagenic pig genes.

## Conclusions

Understanding the collagen-related genes of *Kele pig* is helpful for us to further explore the regulatory factors of the collagen-related traits of *Kele pig*. In this study, a total of 307 candidate genes related to collagen traits were excavated, including *COL9A1*, *COL6A5*, *COL4A4*, *COL4A3*, *EP300*, *SOS2*, *EPO*, etc. By RT-qPCR analysis, we determined the expression levels of four candidate genes, *COL9A1*, *COL6A5*, *COL4A4* and *COL4A3*, in different tissues of *Kele pigs*, among which two genes, *COL9A1* and *COL6A5*, were significantly expressed in the skin tissues of *Kele pigs*. Both *COL4A4* and *COL4A3* genes were significantly expressed in the kidney tissue of *Kele pigs*. The specificity of 14 sites randomly selected from related genes was consistent with the results of whole genome resequencing. These results indicate that the specific SNP molecular marker information obtained by whole genome resequencing can be used as the basis for the analysis of collagen traits in *Kele pigs*, and these genes may be potential targets

for domestication, reproduction and selection in the past and in the future. The results of this study are helpful for further research on the regulation of collagen traits and the development and utilization of *Kele pigs*.

## Supporting information

**S1 File.**
(ZIP)

## Author Contributions

**Conceptualization:** Wei Chen.

**Data curation:** Yu Dan Zhang, Wei Yuan.

**Formal analysis:** Wei Yuan, Xiao Yang.

**Funding acquisition:** Wei Chen.

**Investigation:** Yu Dan Zhang, Wei Yuan, Huan Bi, Xiao Yang.

**Methodology:** Yu Dan Zhang, Wei Yuan, Huan Bi.

**Project administration:** Wei Chen.

**Resources:** Yi Yu Zhang, Wei Chen.

**Software:** Yu Dan Zhang, Wei Yuan.

**Supervision:** Yi Yu Zhang, Wei Chen.

**Validation:** Yu Dan Zhang.

**Visualization:** Yu Dan Zhang.

**Writing – original draft:** Yu Dan Zhang.

**Writing – review & editing:** Yu Dan Zhang.

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
