## [Decision Letter · Decision Letter 0]

19 Jun 2024

PONE-D-24-18276Whole-Genome Resequencing Reveals Collagen-related genes in Kele pigsPLOS ONE

Dear Dr. Zhang,

Thank you for submitting your manuscript to PLOS ONE. After careful consideration, we feel that it has merit but does not fully meet PLOS ONE’s publication criteria as it currently stands. Therefore, we invite you to submit a revised version of the manuscript that addresses the points raised during the review process.

We look forward to receiving your revised manuscript.

Kind regards,

Rajesh Kumar Pathak, Ph.D.

Academic Editor

PLOS ONE

“This research was supported by the Guizhou Provincial Science and Technology Program (Guizhou Kehe Support [2022] General 087) and the Guizhou University Talent Introduction Scientific Research Project (Guida Daren Jihe Zi [2019] No. 34).”

Additional Editor Comments:

Authors are required to revise the manuscript based on the reviewers' comments. Additionally, please ensure that a comprehensive biological interpretation of your findings is provided. The Discussion section must be updated, and significant revisions are needed throughout the manuscript.

Reviewers' comments:

Reviewer's Responses to Questions

**Comments to the Author**

1. Is the manuscript technically sound, and do the data support the conclusions?

Reviewer #1: Partly

Reviewer #2: Partly

Reviewer #3: Yes

Reviewer #4: Partly

Reviewer #5: No

2. Has the statistical analysis been performed appropriately and rigorously? 

Reviewer #1: Yes

Reviewer #2: Yes

Reviewer #3: I Don't Know

Reviewer #4: No

Reviewer #5: Yes

3. Have the authors made all data underlying the findings in their manuscript fully available?

Reviewer #1: Yes

Reviewer #2: No

Reviewer #3: Yes

Reviewer #4: Yes

Reviewer #5: No

4. Is the manuscript presented in an intelligible fashion and written in standard English?

Reviewer #1: Yes

Reviewer #2: No

Reviewer #3: Yes

Reviewer #4: No

Reviewer #5: No

5. Review Comments to the Author

Reviewer #1: Recommendation: Major and Minor revisions

Comments:

The topic is very interesting and the experiments were carefully performed by authors. This article highlights several genes that could be considered potential candidates for encoding collagen proteins in future research.

Thus, I recommend this article for publication in PLOS ONE after the following major and minor points are addressed by the authors.

Major Revision:

1. Authors' results indicated that the expression levels of COL9A1, COL6A5, COL4A4 and COL4A3 were investigated using RT-qPCR but are there any other supporting experiments done to confirm this overexpression of these genes?

Minor Revision:

1. What is the exact concentration of genomic DNA used in agarose gel electrophoresis?

2. In this manuscript some DNA samples A260/A280 is near to 2, but generally A260/A280 should be in the range of 1.8–1.9.

3. Except the fine meat quality and excellent flavor, are there any significant reasons to choose Kele pig as a model?

4. The conclusion is not impressive in its current form. Conclusion section of the manuscript requires further development to strengthen its impact and clarity.

5. It is recommended to include references to the recently published manuscripts on collagen-related genes to ensure the manuscript reflects the latest advancements in the field.

6. Authors should ensure that the manuscript is free from typos and grammatical errors.

Reviewer #2: 1. I did not find it relevant to utilize SNP/INDEL extraction from whole genome resequencing data to reveal the collagen related genes in Kele pigs from different tissues as it should have done using RNA-seq data as you are validating the change in level of expression using RT-PCR. Objective and conclusion of your study is failed to explain utilization of SNP/INDEL from whole genome resequencing data in place of RNA-seq data, which is the standard approach to access the expression of genes. Relevant explanation of it should be added and how its better to use SNP/INDEL extraction from whole genome resequencing data.

2. Title, abstract and headings should be improved.

3. There is no methodology provided for genomic localization (e.g. Exon etc.) of SNP/INDELs. Further, no methodology is provided to get impact of SNP/INDELs and for gene annotation of the SNP/INDELs as directly mentioned the KEGG analysis which is only possible after extraction of candidate genes of SNP/INDELs.

4. No detail of submission of raw sequencing data to NCBI.

5. Quality of writing could be improved.

Reviewer #3: Dear Zhang et al.,

I have following comments for your manuscript.

1. Line 111 E.coli should be in italics

2. Line 195, 196, and 197 …what is G ?

3. Linne 210 what is 9.04×.

4. Line 2-4 is not clearly specifying the objective of the study

5. Line 5-7 should be merged

6. Figure 5 labelling on X axis should be done in english

7.

8. Line 183 ..mentioned Figure 1 does not correlate with Fig 1 mentioned

9. Line 185 past tense should be used

10. Line 190 : Title doesn’t seem to be appropriate

11. Define G (Filter number) in line 201

12. Line 213 and 214 should be explained more.

13. Line 216 to Line 222, including Table 4 should mention SNP’s and Indel with respect to each of the genome samples (Kel1, kel2, Kel3, kel4 and kel5)

Reviewer #4: In the manuscript “Whole-Genome Resequencing Reveals Collagen-related Genes in Kele pigs", submitted by Zhang et al., the authors investigate genes involved in collagen-protein traits in Kele pigs and suggest a potential tissue-specific regulation of these collagen genes.

As a researcher working with Omics, I found substantial innovation in the approach to developing the data and its usefulness for a wide range of readers. The manuscript may be recommended for publication in its present form. Nevertheless, the authors may be asked to look into the following major editorial corrections:

• The objective is clearly stated, focusing on the identification and analysis of collagen-related genes in Kele pigs using whole-genome resequencing. However, it would be beneficial to include a brief statement on the significance of collagen-related traits in Kele pigs to contextualize the study's importance.

• The identification of high, medium, and low-impact mutation sites is thorough, with exact numbers provided. However, an explanation of how "impact" is defined and categorized would be beneficial for clarity.

• The choice of tissues (kidney and skin) and the rationale behind selecting these tissues for expression analysis should be explained in more detail.

• The manuscript reports extensive mutation data, including the identification of 4,570 high-impact SNPs, 132,256 medium-impact SNPs, and 318,150 low-impact SNPs. While these numbers are informative, it would be beneficial to include statistical measures such as mutation rates per base pair or per gene to provide a sense of scale and relevance.

• For the INDEL annotation results, similar statistical measures would enhance the understanding of their significance. Including a comparison of observed mutation frequencies against expected frequencies could provide insights into whether certain types of mutations are overrepresented.

• In the RT-qPCR analysis, it is important to report the statistical methods used to assess gene expression differences. Information on replicates, the statistical tests used (e.g., ANOVA, t-tests), and p-values or confidence intervals for the expression levels would strengthen the validity of the findings. Additionally, including measures of variance (such as standard error or standard deviation) for the expression levels would provide a clearer picture of the data's reliability.

• It is recommended to perform multiple testing corrections (e.g., Bonferroni or FDR) for the differential expression analysis to account for the possibility of Type I errors due to multiple comparisons.

• For the KEGG and GO enrichment analyses, details on the statistical thresholds for significance (e.g., p-value cutoff, false discovery rate) should be provided. Including the enrichment scores and the number of genes in each pathway/category would also enhance the interpretability of the results.

• The conclusion would benefit from a discussion on the broader implications of these findings for the breeding, health, and management of Kele pigs, as well as any potential applications in other pig breeds or species.

• To improve the manuscript, consider expanding the discussion on how these genetic discoveries can be applied practically, both within the context of Kele pigs and in a broader agricultural or biomedical context.

• Ensure that all abbreviations (e.g., SNP, INDEL, KEGG, GO) are defined upon first use to aid readers who may not be familiar with these terms.

• Minor grammatical corrections are needed in several areas for better readability. For example, "clean data of 241.04G" should be revised to "241.04 Gb of clean data," and "identified 4570 high-impact mutation sites" should be "identified 4,570 high-impact mutation sites."

• Line 9: "sequencing results found" should be "sequencing results revealed."

• Line 13: "could result in protein loss" should be "could result in protein function loss."

• Line 21: "modified-impact mutation sites" should be clarified or rephrased, as "modified-impact" is not a standard term in genetic annotation.

Reviewer #5: The title of the MS is "Whole-Genome Resequencing Reveals Collagen-related genes in Kele pigs", so it was expected that there will a major focus over collagen related genes and may be ssome new genes or new finding associated with these genes will be revealed, though majority of this was not fulfilled in the MS.

MS focus majorly over the different indels and SNPs and findings of other authors (those with collagen specially).

Biological interpretation of findings is majorly missing.

Whether the data generated is submitted to any public repository, like SRA?

In my opinion, MS needs a major restructuring with proper discussion and interpretation of findings and also focused discussion on "collagen related genes and their finding"

Some general comments are also provided below:

line no 6-9, more seems as results rather than methodology.

Whether line 18-21, "There were 4740 medium-impact mutation sites

19 that have the potential to affect protein properties, as well as 19,298

20 low-impact mutation sites. Furthermore, there were a staggering

21 14,197,763 modified-impact mutation sites identified in the analysis.", is required in abstract?

At line 1191-192, authors mentioned that "After rigorous filtering of the sequencing data, high-quality clean

192 data was obtained.", they should also mention about that how 'rigorous filtering' was carried out

line 198-200 needs restructuring

line 216, "a total of 50,040,8.74 million SNPs loci and" is not clear!, at other places also it is mentioned.

whether in Table 4, mentioning "Kele pig" is needed as all the samples are of that only and table heading also mention it, same in Table 5, Table 6

Line 236, "SNPS"!!

line 270-271, "KEGG enrichment analysis was conducted on genes within the

271 selected regions.", here also provide details about selected regions and their utilities

line 292-"Kore"!

Line 353 : "Li Qiannan's study." reference is missing!

6. PLOS authors have the option to publish the peer review history of their article (what does this mean?). If published, this will include your full peer review and any attached files.

Reviewer #1: **Yes: **Soumi Biswas

Reviewer #2: No

Reviewer #3: No

Reviewer #4: No

Reviewer #5: No

---

## [Author Response · Author response to Decision Letter 0]

6 Aug 2024

Many thanks to all reviewers for their valuable comments and professional comments on this study and on us. We sincerely thank the editors and all reviewers for their feedback, which we will use to improve the quality of the manuscript.We have replied to Reviewers' suggestions one by one in the "Response to Reviewers" file, which has been uploaded. Thanks again to all experts.

---

## [Decision Letter · Decision Letter 1]

30 Aug 2024

PONE-D-24-18276R1Whole-Genome Resequencing Reveals Collagen-related genes in Kele pigsPLOS ONE

Dear Dr. Chen,

Thank you for submitting your manuscript to PLOS ONE. After careful consideration, we feel that it has merit but does not fully meet PLOS ONE’s publication criteria as it currently stands. Therefore, we invite you to submit a revised version of the manuscript that addresses the points raised during the review process.

We look forward to receiving your revised manuscript.

Kind regards,

Rajesh Kumar Pathak, Ph.D.

Academic Editor

PLOS ONE

Additional Editor Comments:

The revised manuscript requires additional refinement to improve clarity and accuracy. Please update the section headings to more accurately reflect their content, especially those related to SNPs, INDELs, and PCR validation. Streamline the Results section into a more cohesive narrative and correct any grammatical errors. Revise the Abstract to clearly summarize the study’s objectives, methods, results, and conclusions. Ensure consistency in reporting sequencing technology and resolve any discrepancies in sequencing depth. Additionally, provide detailed information on data availability and enhance the interpretation of results by connecting them more effectively to broader implications and existing research.

Reviewers' comments:

Reviewer's Responses to Questions

**Comments to the Author**

1. If the authors have adequately addressed your comments raised in a previous round of review and you feel that this manuscript is now acceptable for publication, you may indicate that here to bypass the “Comments to the Author” section, enter your conflict of interest statement in the “Confidential to Editor” section, and submit your "Accept" recommendation.

Reviewer #1: All comments have been addressed

Reviewer #2: All comments have been addressed

Reviewer #5: (No Response)

2. Is the manuscript technically sound, and do the data support the conclusions?

Reviewer #1: Yes

Reviewer #2: Yes

Reviewer #5: Partly

3. Has the statistical analysis been performed appropriately and rigorously? 

Reviewer #1: Yes

Reviewer #2: Yes

Reviewer #5: Yes

4. Have the authors made all data underlying the findings in their manuscript fully available?

Reviewer #1: Yes

Reviewer #2: Yes

Reviewer #5: No

5. Is the manuscript presented in an intelligible fashion and written in standard English?

Reviewer #1: Yes

Reviewer #2: No

Reviewer #5: No

6. Review Comments to the Author

Reviewer #1: All comments have been addressed properly by authors and I think this manuscript contribute significant value to the society.

Reviewer #2: 1. Headings in Experimental samples and methods section such as “Notes on Kele Pig SNPs and INDELs” should be replace with “Assessment of effects of SNPs and INDELs along with extraction of their associated genes”; and

“KEGG and GO annotation and channel analysis in Kele pigs” should be replaced with GO and KEGG analyses of genes associated with SNPs and INDELs”.

2. Additionally the content of heading “KEGG and GO annotation and channel analysis in Kele pigs” should also be improved.

3. There is no need to write results in each sub-heading of Result section.

4. Headings such as SNPs comment result and INDELs Comment Result should be also be improved.

5. Still, there are grammatical mistakes.

6. Abstract should be improved.

7. In Introduction initial lines “Protein is the primary component of organismal structure and serves as the material basis for biological functions and metabolic activities. Its sequence contains a wealth of genetic information [1] .” could be removed as unnecessary.

8. Heading “Validation of specific SNPs sites” could be replaced with “PCR based validation of 14 SNPs and their associated genes”.

9. Heading “Real-time fluorescent quantitative PCR” could be replace with “RT- fluorescent qPCR based validation of role of SNP associated genes in collagen expression”.

10. Headings in Result section should be improved.

Reviewer #5: Manuscript needs a thorough modification as the work done is very interesting and of interest of wider community. Though the results are not properly presented and also not fully interpreted.

Also, I was not able to identify any information about the availability of raw or processed data generated in this manuscript.

Some general comments from my side are:

line 177 : In order to verify the specific SNP sites of red deer ??

line 181-182: According to the sequences of 100 bp before and after SNP sites were selected, according to what ??

line 259-260: Additionally, the average sequencing depth of Kele pig samples was determined to be 9.04×.Initially, in materials and methods, it was mentioned as 10 X!!

line 275 SNPs comment result, is it an appropriate heading ??

line 302: INDELs Comment Result , again !!!

line 408-409: "We used sanger sequencing technology to verify the specific loci of Kele pigs previously screened.", at line 136, Illumina is mentioned while at line 191-192, nothing is mentioned about sequencing technology!!

line 485: In this study, we reported the whole genome resequencing of five Kele pigs. authors are mentioned about whole genome resequencing but not much matter associated with that is available in the MS.

7. PLOS authors have the option to publish the peer review history of their article (what does this mean?). If published, this will include your full peer review and any attached files.

Reviewer #1: No

Reviewer #2: **Yes: **Kalpana Singh

Reviewer #5: No

---

## [Author Response · Author response to Decision Letter 1]

3 Sep 2024

Many thanks to all reviewers for their valuable comments and professional opinions on this study and us again. We sincerely thank the editors and all reviewers for their feedback and we will use this detailed feedback to improve the quality of the manuscript.

---

## [Decision Letter · Decision Letter 2]

19 Sep 2024

Whole-Genome Resequencing Reveals Collagen-related genes in Kele pigs

PONE-D-24-18276R2

Dear Dr. Chen,

We’re pleased to inform you that your manuscript has been judged scientifically suitable for publication and will be formally accepted for publication once it meets all outstanding technical requirements.

Kind regards,

Rajesh Kumar Pathak, Ph.D.

Academic Editor

PLOS ONE

Additional Editor Comments (optional):

The authors have thoroughly addressed all concerns raised by the reviewers and revised the manuscript in accordance with their comments and suggestions. Therefore, I recommend it for acceptance.

Reviewers' comments:

Reviewer's Responses to Questions

**Comments to the Author**

1. If the authors have adequately addressed your comments raised in a previous round of review and you feel that this manuscript is now acceptable for publication, you may indicate that here to bypass the “Comments to the Author” section, enter your conflict of interest statement in the “Confidential to Editor” section, and submit your "Accept" recommendation.

Reviewer #2: All comments have been addressed

Reviewer #6: All comments have been addressed

2. Is the manuscript technically sound, and do the data support the conclusions?

Reviewer #2: Yes

Reviewer #6: Yes

3. Has the statistical analysis been performed appropriately and rigorously? 

Reviewer #2: Yes

Reviewer #6: Yes

4. Have the authors made all data underlying the findings in their manuscript fully available?

Reviewer #2: Yes

Reviewer #6: Yes

5. Is the manuscript presented in an intelligible fashion and written in standard English?

Reviewer #2: Yes

Reviewer #6: Yes

6. Review Comments to the Author

Reviewer #2: (No Response)

Reviewer #6: The author has effectively addressed all the queries raised in the previous review. All aspects highlighted in the feedback have been carefully considered and effectively incorporated into the revised manuscript. Based on these improvements and the overall quality of the revised work, I recommend the manuscript for publication.

7. PLOS authors have the option to publish the peer review history of their article (what does this mean?). If published, this will include your full peer review and any attached files.

Reviewer #2: **Yes: **Kalpana Singh

Reviewer #6: No

---

## [Editor Report · Acceptance letter]

1 Oct 2024

PONE-D-24-18276R2 

PLOS ONE

Dear Dr. Chen, 

I'm pleased to inform you that your manuscript has been deemed suitable for publication in PLOS ONE. Congratulations! Your manuscript is now being handed over to our production team.

Kind regards, 

on behalf of

Dr. Rajesh Kumar Pathak 

Academic Editor

PLOS ONE